# Easing Non-IID Pain with Dual Relaxations in Federated Learning: SimFAFL redeems an enhanced efficacy

## Abstract

Federated learning enables collaborative training across clients without sharing local data, making it well-suited for privacy preservation. However, statistical heterogeneity in local datasets, known as non-identically distributed (non-iid) problem, leads to client drift and poor convergence for global model. Feature alignment for federated learning (FAFL) methods have emerged to tackle this problem by constraining local feature distribution with global per-class representations and achieved remarkable performance. However, issues persist around 1) lacking expandability and extensibility (i.e., tight coupling with classification tasks), 2) requiring additional communication and computational cost and 3) expecting rigorous theoretical analysis. To address these issues, this paper presents a simpler version of FAFL - SimFAFL, which decouples the constantly updated FAFL constraints from explicit categorical dependencies using two modular constraints. Specifically, the proposed constraints are i) a fixed global reference distribution and ii) globally shared task parameters. These act as centroid and shape regularizers to restrict drift in local feature distributions without requiring explicit categorization. We provide theoretical analysis proving the constraints reduce the deviation upper bound of the objective function, demonstrating efficacy of SimFAFL in mitigating harmful drift. Extensive experiments demonstrate SimFAFL's state-of-the-art performance compared to prevalent methods. Moreover, the modular design also expands model flexibility and benefits generalization without imposing communication/computation costs.

## 1 Introduction

Federated learning (FL) has emerged as a distributed machine learning approach that enables collaborative model training across clients, without the need to directly share local private data. This allows organizations to construct collective intelligence while adhering to stringent data privacy regulations. The horizontal federated optimization process typically involves clients' local training (i.e., alternating rounds of localized stochastic gradient descent on each client's data), followed by server aggregation (i.e., weighted aggregation of local models on the central server) (McMahan et al. (2017)). However, a persistent challenge is that real-world client datasets often demonstrate statistical heterogeneity, with each client exhibiting distinct data distribution patterns (Zhao et al. (2018)). This non-identical distribution of data across devices, known as the **non-IID problem** (non-identically independent distributed) in federated learning literature, can cause client drift and impede model convergence. The main issue lies on the divergence in client data distributions during localized training, leading to personalized drift in gradient updates, which degrades accuracy when aggregating to the global. To mitigate this issue, personalized federated learning (PFL) techniques have been proposed (Arivazhagan et al. (2019), Collins et al. (2021), Li et al. (2021)). PFL aims to adapt the personalized models to local data distribution, while benefits from cooperative training. Some data-based PFL methods attempt to reduce data divergence among clients by data augmentation or data generation, relying on proxy datasets (Zhao et al. (2018)). While model-based PFL methods introduce regularization to reduce client drift during local training without the need of proxy datasets (Tan et al. (2022), Chen & Chao (2021), Karimireddy et al. (2020)). Feature alignment is a typical model-based regularization, which has achieved remarkable results.

Feature Alignment for Federated Learning (FAFL) methods have emerged to improve performance under heterogeneous data distributions ( Xu et al. (2023), Zhang et al. (2023), Tan et al. (2022)). These techniques introduce regularization terms during training to align local feature representations with global per-class centroids. By constraining divergence in latent space, FAFL has been shown to mitigate harmful client drift and improve convergence compared to unregularized federated optimization.

However, current FAFL approaches still have certain limitations. First, the reliance on explicit per-class representations reduces extensibility to non-classification tasks, such as regression task. Without predefined categories, the required global centroids cannot be obtained. Second, existing techniques expect rigorous theoretical analysis justifying their particular constraints and proving superior convergence. Regrettable, FedPAC ( Xu et al. (2023)) nor FedCR ( Zhang et al. (2023)) have not done such work. Finally, continuous aggregation of sample features on the server for up-to-date centroids incurs additional communication and computation costs, which we will detail in Section 2.

To address these issues, we first provide formal convergence analysis of FedPAC ( Xu et al. (2023)), a prototypical FAFL algorithm. Drawing on Li et al. (2019), we prove FedPAC's feature alignment constraints provably reduce the theoretical upper bound on loss deviation between federated and centralized learning. This offers a principled explanation of how FAFL mitigates drift compared to FedAvg. Furthermore, we identify centroid and shape drift as key factors causing performance degradation under non-IID data, which FAFL aims to restrict such clients' drift.

Table 1: Test accuracy (%) of Pre-experiments.

| Methods | CIFAR10-1 | CIFAR10-2 | CIFAR100-1 | CIFAR100-2 |
|---------|-----------|-----------|------------|------------|
| FedAvg | 76.86 | 76.07 | 26.87 | 28.54 |
| FedCR | 81.55 | 82.09 | 59.06 | 42.91 |
| Pre-exp I | 81.42 | 81.48 | 57.72 | 41.41 |
| Pre-exp II | 84.44 | 83.50 | 57.39 | 38.16 |

Motivated by these insights, we propose a simplified and extensible FAFL framework called Sim-FAFL that relaxes the strict alignment constraints. To avoid continuous centroid updates, we constrain local features to a fixed global reference distribution, which improves accuracy as shown in Table 1. However, this only aligns centroids. To further capture shape, we freeze global task parameters from the previous round for local training. Moreover, as analyzed in Section 4.3 and 5, both constraints provably reduce loss divergence. Experiments demonstrate enhanced performance and robustness over existing FAFL approaches, while expanding model capabilities. In addition, ablation experiments show that both constraints provide significant performance improvements individually. Furthermore, we investigate the sensitivity of model accuracy to the constraint coefficient in extension studies.

**Contributions.** This paper elucidates several pivotal contributions:
**1. Convergence Analysis of FAFL**: We present a comprehensive analysis and proof of the convergence of FAFL, underscoring the theoretical utility of global representation constraints.
**2. Introduction of Simplified Constraints**: The ongoing updates in FAFL methods' representations are simplified into 2 unyielding constraints: i) a fixed representation and ii) global task parameters. Note, these newly introduced constraints deftly limit the feature distribution in local training.
**3. Empirical Validation for Enhanced Performance and Robustness**: Rigorous experimental assessments substantiate the enhanced performance and robustness of SimFAFL. This empirical evidence affirms the superiority of the proposed dual constraints, offering practical proof of SimFAFL's broadened applicability and efficacy in diverse tasks.

**Benefits.** The introduction of SimFAFL, as propounded in this work, yields considerable benefits:
**1. Enhanced Performance**: SimFAFL outperforms existing methodologies, particularly in scenarios with heterogeneous client data distributions.
**2. Efficiency and Privacy**: By circumventing the need for computation and transmission of features, SimFAFL not only conserves time and computational resources but also significantly enhances the privacy protection of local data.
**3. Expanded Applicability**: The relaxed constraints, now more generic, find utility in a broader array of tasks beyond classification, including regression tasks.

**4. Robustness and Generalization**: SimFAFL exhibits augmented robustness and generalization capabilities, standing as a testament to its well-rounded design.

The remainder of this paper is organized as follows. Section 2 reviews relevant prior work, including typical FAFL algorithms. Section 3 formulates the problem and analyzes the convergence and intuitions behind FAFL. Section 4 further introduces the proposed SimFAFL method and provides theoretical efficacy analysis. Section 5 presents our experimental setup, results, and discussions. Finally, Section 6 concludes the paper.

## 2 RELATED WORKS

**Feature Alignment for Federated Learning (FAFL).** Recently, feature alignment techniques have gained traction in federated learning as a way to constrain local model adaptations based on globally shared representations. The notion of using prototype features, such as averaged centroids, as a proxy for task classes has been explored across various domains (e.g., FedProto, Tan et al. (2022))). Building on prototype regularization concepts, FedPAC (Xu et al. (2023)) combines explicit alignment of local features to global class centroids along with model averaging. By regularizing local training to remain close to the globally aggregated per-class means, FedPAC induces a form of feature distribution alignment to mitigate client drift. Extending this, FedCR (Zhang et al. (2023)) models the global class representations as Gaussian distributions rather than deterministic points. The mean and variance parameters are then aggregated across clients using a product of experts approach Hinton (2002). Local updates aim to minimize the KL divergence between their learned feature distributions and the global centroids. Since FedPAC uses an L2 loss, it can be viewed as a special case of FedCR's divergence regularization framework. While these methods have empirically demonstrated the benefits of feature alignment, they lack rigorous theoretical analysis into why constraining distributions improves optimization. Our work consequently aims to address this gap by providing formal convergence guarantees that reveal how alignment reduces deviation between local and global objectives. We further distill key principles that motivate a simplified approach without reliance on explicit class representations.

**Convergence Guarantees of non-IID Federated Learning.** While performance degrades under non-IID data, federated optimization can still converge. Formal proofs are provided in several works including Zhao et al. (2018) and Li et al. (2019). Li et al. (2019) established that the model deviation between federated learning and centralized learning diminishes to zero as rounds increase, under common assumptions like learning rate decay. Meanwhile, Zhao et al. (2018) characterized the dependence of this deviation on data heterogeneity, motivating distribution alignment techniques. Related results (e.g., Zhu et al. (2021) and Ben-David et al. (2006)) are cited in FedCR, though lacking detailed discussion ( Zhang et al. (2023)), indicated that the upper bound of loss deviation is positively correlated with divergence of distribution. Ultimately, despite empirical success, FAFL methods are expecting supporting theory regarding the efficacy of their constraints in reducing deviation and improving convergence.

## 3 CONVERGENCE ANALYSIS OF FAFL

Based on the problem of interest, this chapter provides theoretical analysis and proof regarding the convergence properties of feature alignment for federated learning (FAFL) methods. Note, this offers theoretical grounding for the empirical benefits of feature alignment in mitigating heterogeneity.

**Problem Formulation.** We formally define the optimization problem for feature alignment in federated learning. Let $f_i(\phi_i)$ denote client $i$'s feature extractor parameterized by $\phi_i$. Let $v_i$ represent personalized task parameters. Following FAFL conventions, the feature extractor is globally shared while task parameters are localized. The end-to-end model for client $i$ can be expressed as $\mathcal{F}_i(\phi_i, v_i; x_i)$, where $(x_i, y_i)$ denotes the local dataset. The local objective function of client $i$ is

$$\mathcal{L}(\phi_i, v_i; x_i, y_i) = \mathcal{L}_s(\mathcal{F}(\phi_i, v_i; x_i), y_i) + \lambda \mathcal{L}_d(f(\phi_i; x_i), \bar{C}) \qquad (1)$$

where $\bar{C}$ denotes the global per-class centroids aggregated from clients features, $\lambda$ controls the regularization strength, $L_s(\cdot)$ denotes the empirical risk of original task, and $L_d(\cdot)$ denotes the feature

alignment loss between the global per-class centroids and the local centroids. In FedPAC, $L_d(\cdot)$ denotes L2-norm regularizer, while for FedCR, it denotes KL divergence.

The global objective aggregates local objective function weighted by the data fraction $p_i$ at client $i$

$$\mathcal{L}_t^g = \sum_i p_i \mathcal{L}_s(\bar{\phi}, v_i; x_i, y_i) + \lambda \mathcal{L}_d(f(\bar{\phi}; x), \bar{C}_t) \tag{2}$$

where $\bar{\phi}$ represents the globally shared parameters, $p_i$ denotes the ratio of sample number of client $i$ to the global data. The objective function deviation between the model of current communication round $t+1$ and the optimal model can be expressed as

$$\mathcal{L}_{t+1}^g - \mathcal{L}_t^{g*} \leq \frac{L}{2} \parallel \bar{w}_{t+1} - w_t^* \parallel_2 + \lambda \parallel \bar{C}_{t+1} - \bar{C}_t \parallel_2 \tag{3}$$

**Convergence Guarantees.** We analyze the convergence behavior of FAFL methods. Based on standard assumptions Appendix A.1 and theorems from Li et al. (2019), the main difference between the objective function of FAFL and original FL is that FAFL involves updating of $\bar{C}$, making loss function change during training. The upper of loss deviation represented as $\triangle_{t+1} = \frac{L}{2} \parallel \bar{w}_{t+1} - w_t^* \parallel_2 + \lambda \parallel \bar{C}_{t+1} - \bar{C}_t \parallel_2$ follows $\triangle_{t+1} \leq (1 - \eta_t \mu)\triangle_t + \eta_t^2 D$, where $\eta_t$ denotes the learning rate of round $t$. Thus, we can show that the gap between the global objective $\mathcal{L}_g$ and optimal centralized model $\mathcal{L}_g^*$ satisfies

$$\mathbb{E}\left[\mathcal{L}_{t+1}^g - \mathcal{L}_t^{g*}\right] \leq \frac{\kappa}{\gamma + t}\left(\frac{2H}{\mu} + \frac{\mu(\gamma + 1)}{2}\mathbb{E} \parallel \bar{w}_1 - w^* \parallel_2^2\right) \tag{4}$$

Note, this demonstrates that FAFL achieves a convergence rate of $\mathcal{O}(\frac{1}{T})$ despite non-IID data, akin to FedAvg. The full proof is provided in the Appendix A.1.

**Efficacy of Feature Alignment.** This subsection analyzes how feature alignment improves convergence. Based on the assumptions in Appendix A.1 and analysis in Tan et al. (2022), the upper loss deviation can be bounded as

$$\mathcal{L}_{t+1}^g - \mathcal{L}_t^g \leq (\frac{L}{2}\eta^2 - \eta) \sum_i p_i \sum_e \parallel \nabla^i(\bar{w}_{tE+e+\frac{1}{2}}) \parallel_2^2 + \frac{1}{2} \sum_i p_i \parallel \nabla^i(w_{(t+1)E+\frac{1}{2}}^i) \parallel_2^2$$
$$+ \frac{1}{2} \sum p_i \parallel \bar{w}_{t+1} - w_{t+1}^i \parallel_2^2 + \frac{L}{2}E\eta^2\sigma^2 - \lambda \parallel \bar{C}_{t+1} - \bar{C}_{t+2} \parallel_2 \tag{5}$$

which can be transferred in the format

$$\mathcal{L}_{t+1}^g - \mathcal{L}_t^g \leq \frac{1}{2}G^2 + 2\eta_t^2(E-1)^2G^2 + \frac{L}{2}\eta_t^2\sigma^2 - \lambda \parallel \bar{C}_{t+1} - \bar{C}_{t+2} \parallel_2 \tag{6}$$

where the $\lambda$-dependent term can be denoted as

$$T(\lambda) = (4\eta_t^2(E-1)^2 + 1)G_2^2\lambda^2 + L\eta_t^2\sigma_2^2\lambda^2 - \lambda \parallel \bar{C}_{t+1} - \bar{C}_{t+2} \parallel_2 \tag{7}$$

This indicates that $T(\lambda)$ is a quadratic function to $\lambda$ with $T(0) = 0$ that opens upwards. Therefore, there should exist a positive interval where $T(\lambda) < 0$, reducing the overall upper bound. Intuitively, this reveals the FAFL regularization term can decrease loss divergence when properly tuned. The detailed derivations of Eq 5- 7 are provided in the Appendix A.1.

**How FAFL works: an intuitive visualization.** To provide intuition for how FAFL improves federated learning, we visualize the latent feature distributions from two different FedCR clients on the CIFAR-10 dataset, shown in Figure 1. The left plots show the client feature distributions (i.e., client1 and client 6 in round 101) after local training without any alignment constraints. We observe clear centroid and shape divergence between the two clients, seen in the shifting of color-coded class clusters like orange and blue. Consequently, FAFL introduces global class representations during local training to restrict this harmful drift. As illustrated in the right plots, constraining features to public centroids maintains more consistent distributions across clients. Overall, the global class representations act to align the centroid and shape of local features. This mitigates the impact of non-IID data distributions.

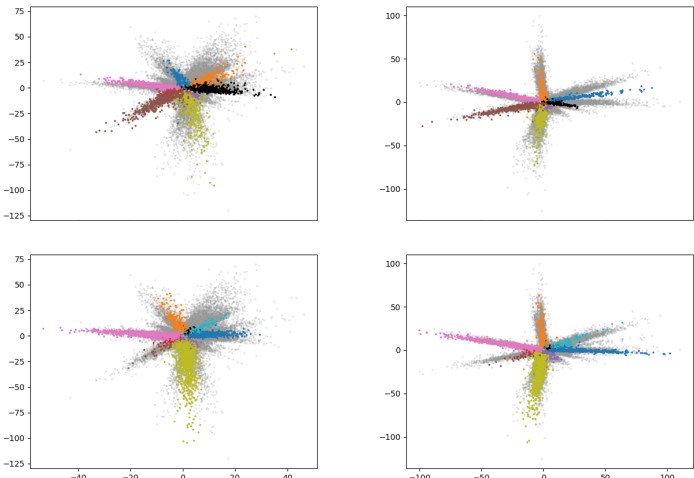

Figure 1: Two-dimensional latent feature distributions for different FedCR clients on CIFAR-10. Without alignment (left), we see centroid and shape divergence across clients. The proposed constraints (right) reduce this drift. Features are color-coded by ground truth class.

## 4 PROPOSED SIMFAFL

While feature alignment improves federated learning, prevailing techniques like FedPAC and FedCR have limitations in extensibility and efficiency. To address these issues, we propose SimFAFL, which relaxes the strict alignment constraints into two modular components.

### 4.1 FIXED GLOBAL REPRESENTATION CONSTRAINTS

To make the constraints independent of the class distribution, we attempt to calculate one global representation rather than per-class representations. A straightforward method to obtain an overall representation for feature alignment is to aggregate all features of all clients into one centroid. Since FedPAC ( Xu et al. (2023)) is a special case of FedCR ( Zhang et al. (2023)), we employ the more general FedCR as the basic framework. Base on FedCR, we perform feature distribution aggregation on features of all samples by PoE ( Hinton (2002)). To preliminarily verify the feasibility of this novel constraint, we conducted two pre-experiments with FedAvg ( McMahan et al. (2017)) and FedCR as baseline. Pre-experiment I implements feature alignment with the continuously updated overall centroid. Although the overall representation can be decoupled from the classification task, additional calculations are still required due to continuous updates. Therefore, we conduct pre-experiment II, which enforces a fixed feature distribution (multi-dimensional standardized normal distribution) as a global representation. The pre-experiments are conducted on two datasets (CIFAR10 and CIFAR100) with two data partition methods (non-IID1 and non-IID2), which will be intorduced in Section 5.

Table 1 shows that the fixed distribution constraint achieves better accuracy than baselines and pre-experiment I on CIFAR10. However, that the improvement of pre-experiment I and pre-experiment II compared to FedAvg is lower than FedCR on CIFAR100. This is reasonable because the two improved constraints only restrict the feature centroid drift, whereas FAFL constrained the centroid and shape drift simultaneously. The pre-experiments verify the validity of the fixed global representations $z_f$, and this constraint on centroid alignment can be expressed as

$$\mathcal{L}_{ca}^i = \mathcal{L}_d(f_i(\phi_i; x_i), z_f) \tag{8}$$

## 4.2 Global task layer parameter constraints

To restrict the task-dependent feature distribution shapes, we consider the global task branch as a constraint condition. Specifically, we consider the network for a specific task as the feature extractor and the task-related branch. Structurally, the feature extractor can be adapted to other tasks while the task branch can only be applied to the specific task. During local training, the task branches' drift occurs due to data distribution heterogeneity among clients. The drift of the task branch further leads to the drift of the feature distribution generated by the feature extractor, as shown in Figure 1. To mitigate the distribution drift, we introduce a common frozen task branch into the local training process. The task branch $v_f$ is aggregated from the local branches $v_i$ of each client in the last round. This constraint on alignment of distribution shape can be expressed as

$$\mathcal{L}_{sa}^i = \mathcal{L}_s(\mathcal{F}(\phi_i, v_f; x_i), y_i) \tag{9}$$

## 4.3 Analysis and Discussion

**Limitations.** While the proposed method demonstrates strong performance under heterogeneous conditions, the loss deviation in Eq. 4 indicates there may be even greater gains in extreme non-IID scenarios. This suggests our approach may not be uniformly optimal across all data distributions. Additionally, although SimFAFL avoids continuous alignment communication, the proposed constraints do introduce some extra computational overhead during local training.

**Key Insights.** Constraining all feature distributions to a common centroid may seem counterintuitive, as it could overly restrict the classifier. However, experiments show accuracy improvements, likely because compacting features provides more difficult samples to improve decision boundaries. The dense clusters also indirectly enhance the local task network's capabilities. Furthermore, prior techniques like FedCR perform aggregation concurrently with unstable local training, risking inaccurate centroids which may degrade performance.

Overall, based on observed feature drift patterns and analysis of prior alignment techniques, we propose two improved constraints: 1) a fixed reference distribution to constrain local feature centroids, and 2) frozen global task parameters to restrict feature distribution geometry. The local objective (Eq. 10) formalizes how SimFAFL elegantly incorporates these constraints to bound drift while avoiding continuous alignment communication and reliance on task categories.

$$\mathcal{L}^i = \mathcal{L}_s^i(\phi_i(\theta(x)), y) + \beta_1 \mathcal{L}_{ca}^i + \beta_2 \mathcal{L}_{sa}^i \tag{10}$$

where $\beta_1$ and $\beta_2$ denote the coefficients of the two constraint terms, respectively.

# 5 Experiments

We empirically evaluate SimFAFL against state-of-the-art federated learning algorithms, assessing convergence, accuracy, and robustness on benchmark datasets.

## 5.1 Experiment settings

**Datasets.** We train and evaluate the proposed SimFAFL and other methods on four datasets, i.e., EMNIST-L, Fashion-MNIST (FMNIST), CIFAR10, and CIFAR100.There are two common methods for data partition employed for non-IID settings. The first one is that each client of EMNIST-L, and CIFAR10 is randomly assigned 5 classes(3 classes per client for FMNIST and 15 classes per client for CIFAR100), with the same amount of data for each class; The second one is that each client has an undetermined number of classes and the sample labels of clients is set according to a Dirichlet distribution with a Dirichlet parameter 0.5 for EMNIST-L, FMNIST and CIFAR10 (for CIFAR100, Dirichlet parameter is set as 0.3). All data is split into 70% training set and 30% evaluation set, which have the same data distribution. These two non-IID settings are represented as **non-IID1** and **non-IID2**.

**Baselines.** We compare average accuracy of all clients on the evaluation sets of the proposed Sim-FAFL to other methods, with fine-tuning version denoted as "-FT". These methods include FedAvg ( McMahan et al. (2017)), FedSR ( Nguyen et al. (2022)), FedPer ( Arivazhagan et al. (2019)), FedRep ( Collins et al. (2021)), LG-FedAvg ( Liang et al. (2020)), FedBABU ( Oh et al. (2021)),

Table 2: Final test accuracy (%) of the proposed SimFAFL and other methods.

| Method | EMNIST | | FMNIST | | CIFAR10 | | CIFAR100 | |
|---|---|---|---|---|---|---|---|---|
| | non-IID1 | non-IID2 | non-IID1 | non-IID2 | non-IID1 | non-IID2 | non-IID1 | non-IID2 |
| FedAvg | 94.3988 | 94.5000 | 84.1754 | 86.6588 | 76.8556 | 76.0722 | 26.8667 | 28.5389 |
| FedAvg-FT | 96.2500 | 95.8036 | 95.8294 | 91.7630 | 83.6444 | 83.3500 | 53.1222 | 38.3222 |
| FedPer | 94.21429 | 91.39286 | 95.08531 | 88.3223 | 71.2278 | 68.8056 | 39.4944 | 24.3167 |
| LG-FedAvg | 87.8571 | 85.1429 | 94.0569 | 87.1043 | 62.9056 | 63.7611 | 43.9889 | 27.3722 |
| FedRep | 91.1369 | 87.8333 | 93.0995 | 85.9479 | 67.7333 | 67.4167 | 40.6444 | 23.1667 |
| FedBABU | 94.2143 | 91.3929 | 95.0569 | 88.3460 | 71.2278 | 68.8056 | 39.4944 | 24.3167 |
| Ditto | 95.7321 | 95.88010 | 91.1422 | 84.8673 | 78.5111 | 78.9722 | 26.5500 | 35.7833 |
| FedSR-FT | 85.0721 | 80.5706 | 91.9182 | 83.1364 | 57.5833 | 60.5778 | 37.9445 | 24.0167 |
| FedPAC | 97.3512 | 96.6012 | 95.7630 | 91.5024 | 82.9611 | 81.2167 | **63.7000** | 41.1444 |
| FedCR | 97.1765 | 96.3088 | 96.3318 | **93.0000** | 82.9161 | 82.7333 | 59.0556 | 42.9111 |
| **SimFAFL (Ours)** | **97.6824** | **97.0015** | **96.7400** | 92.8227 | **85.0722** | **84.2222** | 61.2111 | **46.1333** |

Ditto ( Li et al. (2021)), FedPAC ( Xu et al. (2023)) and FedCR ( Zhang et al. (2023)). Note, the experimental settings on datasets and comparative methods are similar with settings in FedCR, for convenient comparison.

**Model architecture.** All methods share the same model architecture per dataset - fully-connected nets for EMNIST and CNNs for FMNIST, CIFAR10, CIFAR100 datasets. The CNN extractor has two convolutional layer, two max-pooling layers and two fully-connected layers, while two fully-connected layers correspondingly for EMNIST.

**Hyperparameters.** We use 100 clients with 0.1 participation per round. Constraint coefficients are $\beta_1 = 0.0025$, $\beta_2 = 0.1$ for 10-class tasks (EMNIST, FMNIST and CIFAR10), and $\beta_1 = 0.003$, $\beta_2 = 0.2$ for CIFAR-100.

## 5.2 COMPARISON TO STATE-OF-THE-ART METHODS

The performance of the proposed SimFAFL approach is benchmarked against state-of-the-art federated learning algorithms under two non-IID settings, as summarized in Table 2. The results demonstrate that SimFAFL achieves the best or second-best performance across the evaluated datasets and heterogeneity conditions. For instance, on the CIFAR-100 dataset under non-IID scenario 2, Sim-FAFL attains an accuracy improvement of 3.2% compared to FedCR. These gains can be attributed to the joint centroid and geometry alignment constraints employed in SimFAFL, which serve to mitigate harmful client drift and improve generalization of localized learning.

As illustrated qualitatively in Figure 1, the proposed technique encourages more consistent feature distributions across heterogeneous clients compared to unconstrained federated learning. SimFAFL also induces more compact intra-class feature distributions while enlarging inter-class margins, facilitating the learning of clear decision boundaries. The convergence plots in Figure 2 further showcase that SimFAFL provides faster convergence and maintains performance gains in later training rounds, which we posit is due to the globally aggregated task parameters being updated continuously throughout the training process.

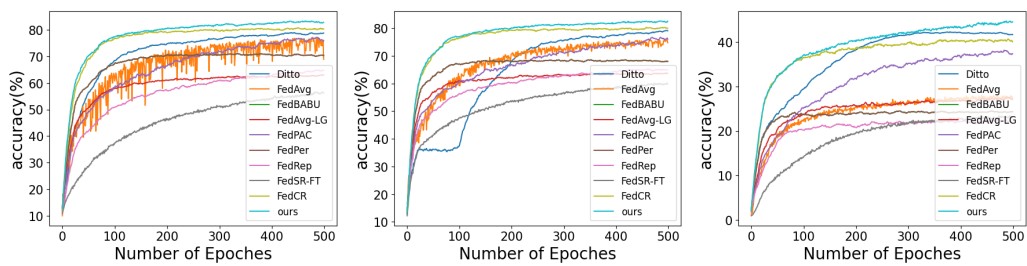

Figure 2: Convergence curves of different methods. The images from left to right show the convergence curves on CIFAR10 with non-IID1, CIFAR10 with non-IID2 and CIFAR100 with non-IID2.

It should be noted that the results in Table 2 do not indicate the optimal accuracy, as the parameters are set in a uniform manner (i.e., fixed settings of $\beta_1 = 0.0025$ and $\beta_2 = 0.1$ for the 10-class tasks (EMNIST, FMNIST and CIFAR10) and $\beta_2 = 0.2$ for CIFAR100). Thus, fine-tune mechanism can help to capture better results. Note, the sensitivity analysis in Section 5.4 demonstrates opportunities for further performance tuning through careful calibration of the regularization weights.

Nonetheless, under uniform settings, SimFAFL demonstrates state-of-the-art capabilities in mitigating non-IID drift.

## 5.3 ABLATION STUDIES

We performed ablation experiments with the two constraints proposed. Specifically, we implement four types of conditions on different data cases, i.e., 1) training without any constraints, 2) training with $\mathcal{L}_{ca}$ for centroid alignment of feature distribution, 3) training with $\mathcal{L}_{sa}$ for shape alignment of feature distribution and 4) training with $\mathcal{L}_{ca}$ and $\mathcal{L}_{sa}$. These four conditions correspond to rows 2 to 5 of Table 3, indicating that both $\mathcal{L}_{ca}$ and $\mathcal{L}_{sa}$ have a positive effect on feature alignment. The combination of two constraints has a better effect than the single constraint, which is reflected in the fact the fourth condition 4) outperformes the other three conditions on each dataset, especially on CIFAR100. In the ten-classification tasks (EMNIST, FMNIST and CIFAR10), the performance gap between single constraint and the joint constraint is slight. The reason is that for simple tasks (CIFAR10), the client drift of feature distributions is not as great as for complex tasks (CIFAR100). Centroid alignment or distribution shape alignment is sufficient to solve the heterogeneity of data distribution for simple tasks. While the CIFAR100 is more difficult, and most methods have not achieved obvious performance on this case. The improvement of performance on CIFAR100 shows that our approach is more advantageous in difficult cases.

Table 3: Average test accuracy (%) on ablation studies of the proposed constraints.

| Method | EMNIST | | FMNIST | | CIFAR10 | | CIFAR100 | |
|---|---|---|---|---|---|---|---|---|
| | non-IID1 | non-IID2 | non-IID1 | non-IID2 | non-IID1 | non-IID2 | non-IID1 | non-IID2 |
| non-constraint | 94.8389 | 94.9533 | 93.3728 | 90.9202 | 63.2619 | 68.3190 | 39.9522 | 29.7148 |
| with $\mathcal{L}_{ca}$ | 96.1380 | 96.3144 | 96.0014 | 92.5024 | 84.4444 | 83.4911 | 57.3922 | 38.1615 |
| with $\mathcal{L}_{sa}$ | 96.2799 | 96.5000 | 95.1832 | 91.3411 | 82.2167 | 82.0900 | 56.6933 | 36.2740 |
| SimFAFL | 97.6824 | 97.0015 | 96.7400 | 92.8227 | 85.0722 | 84.2222 | 61.2111 | 46.1333 |

## 5.4 EXTENSION STUDIES ON HYPERPARAMETER ANALYSIS

We conduct extension studies for the influence of hyper-parameters $\beta_1$ and $\beta_2$ on each dataset with two non-IID settings. Table 4 shows partial results with different hyper-parameters on CIFAR100 with non-IID2 setting. As $\beta_1$ ($\beta$ in FedCR) grow from 0.0001 to 0.008, accuracy of FedCR rises from 40.81% to 42.91% and then down to 14.75%. While the accuracy of SimFAFL rises to 46.78% and then remains above 45%. As $\beta_2$ grows from 0 to 0.6, the accuracy stays between 44% and 47%. $\beta_2$ is fixed at 0.3 when $\beta_1$ varies, and $\beta_1$ is fixed at 0.003 when $\beta_2$ varies. The results indicate that the proposed constraint terms are less sensitive to the value of coefficient. The less sensitivity brings greater robustness in practical. Another rule is reflected in Table 5-7, as $\beta_1$ increases, $\beta_2$ at optimal performance also increases. That is, the simultaneous change of $\beta_1$ and $\beta_2$ in the same direction can lead to better performance. After experiments on various data sets, the optimal parameters in each case are determined. The best $\beta_1$ is 0.0025 for CIFAR10/EMNIST, 0.002 for FMNIST and 0.003 for CIFAR100. The best $\beta_2$ is 0.1 for FMNIST/CIFAR10, 0.2 for EMNIST and 0.3 for CIFAR100.

Table 4: Average test (%) accuracy on CIFAR100 (non-IID2) under varying weight $\beta_1$ and $\beta_2$.

| Weight | 0.0001 | 0.0005 | 0.001 | 0.0015 | 0.002 | 0.003 | 0.005 | 0.008 |
|---|---|---|---|---|---|---|---|---|
| FedCR-$\beta$ | 40.81 | 40.14 | 42.91 | 41.07 | 39.74 | 35.10 | 21.89 | 14.75 |
| SimFAFL-$\beta_1$ | 39.17 | 41.95 | 44.79 | 45.27 | 46.04 | 46.78 | 46.59 | 45.96 |
| Weight | 0 | 0.05 | 0.1 | 0.2 | 0.3 | 0.4 | 0.5 | 0.6 |
| SimFAFL-$\beta_2$ | 44.79 | 45.02 | 45.50 | 46.13 | 46.78 | 46.66 | 46.33 | 46.80 |

# 6 CONCLUSIONS

In this paper, we propose a simplified feature alignment for federated learning method called Sim-FAFL, which decouples the constantly updated alignment constraints into more flexible restrictions based on dual relaxations. The proposed SimFAFL approach confers benefits in terms of enhanced task extensibility and reduced resource requirements for communication and computation. At the same time, SimFAFL achieves state-of-the-art performance compared to existing federated learning methods on benchmark datasets. Furthermore, we provide theoretical analysis into the convergence and advantages of feature alignment techniques, and formally prove the efficacy of SimFAFL in reducing loss divergence across heterogeneous clients. Overall, SimFAFL offers a lightweight and effective framework for FAFL that does not require continuous alignment communications or reliance on task categorization. The dual relaxation constraints provide provable bounds on model divergence while expanding model flexibility.

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

## APPENDIX A PROOFS OF THEORETICAL RESULTS

### A.1 CONVERGENCE OF FEDPAC

**Problem formulation.** The local objective function of client $i$ can be expressed as

$$\mathcal{L}(\phi_i, v_i; x_i, y_i) = \mathcal{L}_s(F_i(\phi_i, v_i; x_i), y) + \lambda \parallel f_i(\phi_i; x_i) - \bar{C} \parallel_2 \tag{11}$$

where

$$\bar{C} = \sum_{i=1}^{m} q_i C_i \tag{12}$$

$$q_i = \frac{\mid D_i \mid}{\sum_{i=1}^{m} \mid D_i \mid} \tag{13}$$

$$\sum_{i=1}^{m} q_i = 1 \tag{14}$$

$$C_i = \frac{1}{\mid D_i \mid} \sum_{D_i} f_i(\phi_i; x_i) \tag{15}$$

The global loss function can be expressed as

$$\mathcal{L}_t^g = \sum_i p_i \mathcal{L}_s(\bar{\phi}, v_i; x_i, y_i) + \lambda \mathcal{L}_d(f(\bar{\phi}; x), \bar{C}_t) \tag{16}$$

**Assumptions 1.** (Lipschitz Smooth). *Each local objective function is L-Lipschitz smooth, which means that the gradient of local objective function is L-Lipschitz continuous,*

$$\mathcal{L}_s^i(w_1) - \mathcal{L}_s^i(w_2) \le \left\langle \triangledown \mathcal{L}_s^i(w_2), (w_1 - w_2) \right\rangle + \frac{L}{2} \parallel w_1 - w_2 \parallel_2^2 \tag{17}$$

**Assumptions 2.** (Convex). *Each local objective function is $\mu$-strongly convex,*

$$\mathcal{L}_s^i(w_1) - \mathcal{L}_s^i(w_2) \ge \left\langle \triangledown \mathcal{L}_s^i(w_2), (w_1 - w_2) \right\rangle + \frac{\mu}{2} \parallel w_1 - w_2 \parallel_2^2 \tag{18}$$

**Assumptions 3.** (Unbiased Gradient and Bounded Variance). *The stochastic gradient $g_i, t = \triangledown \mathcal{L}(w_t, \xi_t)$ is an unbiased estimator of the local gradient for each client. Suppose its expectation*

$$\mathbb{E}_{\xi_i \sim D_i} [g_i, t] = \triangledown \mathcal{L}(w_i, t) \tag{19}$$

and its variance is bounded by $\sigma^2$:

$$\mathbb{E} \left[ \parallel g_i - \triangledown \mathcal{L}^i(w) \parallel^2 \right] \le \sigma^2 \tag{20}$$

**Assumptions 4.** (Bounded Expectation of Euclidean norm of Stochastic Gradients). *The expectation of the stochastic gradient is bounded by $G$*

$$\mathbb{E}[\parallel g_i \parallel_2] \le G \tag{21}$$

**Assumptions 5.** (Lipschitz Continuity). *Each feature extractor is $L_c$-Lipschitz continuous, that is,*

$$\parallel f_i(\phi_{i1}) - f_i(\phi_{i2}) \parallel_2 \le L_c \parallel \phi_i 1 - \phi_i 2 \parallel_2 \tag{22}$$

**Proof of Eq 4.** The main difference between the objective function of FAFL and the normal FL is that FAFL involves updating of $\bar{C}$, which means the loss function changes during training process. Let *Assumption 1* hold and the loss function after round $t + 1$ satisfies

$$\mathcal{L}_{t+1}^g \le \frac{L}{2} \parallel \bar{w}_{t+1} - w_t^* \parallel_2 + \lambda \parallel \bar{C}_{t+1} - \bar{C}_t \parallel_2 \tag{23}$$

where the $\lambda$ term is due to the update of $\bar{C}$. As mentioned in Li et al. (2019), the first term of the upper bound satisfies

$$\| \bar{w}_{t+1} - w_t^* \|_2^2 \leq (1 - \eta_t \mu) \| \bar{w}_t - w_t^* \|_2^2 + \eta_t^2 B \tag{24}$$

where B is a constant related to $\sigma$, $G$, $L$ and the data distribution. Let *Assumption 5* hold, The second term in Eq 23 satisfies

$$
\begin{aligned}
\| \bar{C}_{t+1} - \bar{C}_t \|_2 &\leq \sum_i p_i \| w_{t+1}^i - w_t^i \|_2 \\
&\leq \sum_i p_i (\| w_{t+1}^i - \bar{w}_t \|_2 + \| \bar{w}_t - w_t^i \|_2) \\
&\leq 4\eta_{t+1}^2 (E-1)^2 G^2 + 4\eta_t^2 (E-1)^2 G^2 \\
&\leq 8\eta_t^2 (E-1)^2 G^2
\end{aligned} \tag{25}
$$

where $E$ denotes the number of local steps. The above inequality is derived based on the *Assumption 5* and theorem in Li et al. (2019) that $\| w_{t+1}^i - \bar{w}_t \|_2 \leq 4\eta_{t+1}^2 (E-1)^2 G^2$. The upper bound in Eq 23 can be expressed as

$$
\begin{aligned}
\frac{L}{2} \triangle_{t+1} &= \frac{L}{2} \| \bar{w}_{t+1} - w_t^* \|_2 + \lambda \| \bar{C}_{t+1} - \bar{C}_t \|_2 \\
&\leq \frac{L}{2}(1 - \eta_t \mu) \| \bar{w}_t - w_t^* \|_2 + \frac{L}{2}\eta_t^2 B + 8\eta_t^2 (E-1)^2 G^2 \lambda \\
&= \frac{L}{2}(1 - \eta_t \mu) \| \bar{w}_t - w_t^* \|_2 + \frac{L}{2}\eta_t^2 H \\
&\leq \frac{L}{2}(1 - \eta_t \mu) \| \bar{w}_t - w_t^* \|_2 + \frac{L}{2}\eta_t^2 H + \lambda(1 - \eta_t \mu) \| \bar{C}_{t+1} - \bar{C}_t \|_2 \\
&= \frac{L}{2} \triangle_t + \frac{L}{2}\eta_t^2 H
\end{aligned} \tag{26}
$$

Thus, according to Eq 23 with $\mathcal{L}_{t+1}^g < \frac{L}{2}\triangle_{t+1}$ and $\triangle_{t+1} \leq (1 - \eta_t \mu)\triangle_t + \frac{L}{2}\eta_t^2 H$, the convergence of FAFL can be expressed as

$$\mathcal{L}_{t+1}^g \mathbb{E}\left[\mathcal{L}_t^g - \mathcal{L}^{g*}\right] \leq \frac{\kappa}{\gamma + t}\left(\frac{2H}{\mu} + \frac{\mu(\gamma + 1)}{2}\mathbb{E} \| \bar{w}_1 - w^* \|_2^2\right) \tag{27}$$

where $\kappa$ denotes $\frac{L}{\mu}$ and $\gamma > 0$. This demonstrates that FAFL achieves a convergence rate of $\mathcal{O}(\frac{1}{T})$.

## APPENDIX B    ADDITIONAL EXPERIMENT RESULTS

### B.1    EXTENSION EXPERIMENTS ON $\beta_1$ AND $\beta_2$

In the Tables 5-7, horizontal parameters represent the change of A, and vertical parameters represent the change of B

Table 5: Average test (%) accuracy on CIFAR100 (non-IID1) under varying weight $\beta_1$ and $\beta_2$.

| Weight | 0.001 | 0.0015 | 0.002 | 0.0025 | 0.003 |
|---|---|---|---|---|---|
| **0.1** | 59.75 | 59.99 | 60.13 | 60.22 | 60.73 |
| **0.2** | 59.91 | 60.19 | 60.57 | 60.94 | 61.21 |
| **0.3** | 59.89 | 59.99 | 60.87 | 60.60 | 60.53 |
| **0.4** | 59.68 | 59.74 | 60.97 | 61.06 | 61.42 |

Table 6: Average test (%) accuracy on CIFAR10 (non-IID1) under varying weight $\beta_1$ and $\beta_2$.

| Weight | 0.001 | 0.0015 | 0.002 | 0.0025 | 0.003 |
|---|---|---|---|---|---|
| **0.0** | 84.44 | 84.51 | 84.83 | 84.79 | 84.74 |
| **0.1** | 84.72 | 84.82 | 84.83 | 85.07 | 85.04 |
| **0.2** | 84.57 | 84.74 | 84.87 | 84.86 | 84.87 |

Table 7: Average test (%) accuracy on CIFAR10 (non-IID2) under varying weight $\beta_1$ and $\beta_2$.

| Weight | 0.001 | 0.0015 | 0.002 | 0.0025 | 0.003 |
|---|---|---|---|---|---|
| **0.0** | 83.50 | 84.07 | 83.83 | 83.85 | 83.93 |
| **0.1** | 83.79 | 83.89 | 83.98 | 84.22 | 83.90 |
| **0.2** | 83.63 | 83.65 | 83.65 | 84.02 | 83.77 |
| **0.3** | 83.50 | 83.55 | 83.58 | 83.88 | 83.91 |

