# OpenReview forum: "Easing Non-IID Pain with Dual Relaxations in Federated Learning: SimFAFL redeems an enhanced efficacy"
_ICLR.cc/2024/Conference — ICLR 2024 Conference Withdrawn Submission_

### Official Review · Reviewer_bnjG · 2023-10-31

**Soundness:** 3 good
**Presentation:** 3 good
**Contribution:** 2 fair
**Rating:** 6
**Confidence:** 3

**Summary:**

The paper proposes a method to address non-IID problems, specifically without the constraint of a classification task. The approach involves two key steps: firstly, utilizing a fixed global representation as a stable feature distribution to mitigate centroid drift, and secondly, controlling feature distribution shape drift by employing an aggregated task branch. The authors also provide a convergence analysis of the FAFL approach and present experimental results demonstrating the superiority of the suggested method.

**Strengths:**

This paper suggests that feature alignment can effectively eliminate the limitation associated with solely addressing classification problems, thereby broadening the range of potential applications. Furthermore, the designed structure exclusively focuses on a fixed global centroid, reducing the significant communication costs incurred by continuous centroid updates. The
paper also provides a comprehensive convergence analysis of FAFL, which serves as the theoretical foundation for the proposed method, offering substantial support for its effectiveness.
In the analysis and discussion section, the authors identify existing shortcomings in current methods, providing reasonable and persuasive explanations. They also conduct experiments to investigate the constraints, offering insights into the potential reasons behind the method's performance.

**Weaknesses:**

The inclusion of the fixed global representation in the design constraints is to be justified based on experiments comparing feature alignment with both continuously updated overall centroids and fixed centroids. The results of these experiments exhibited varying outcomes depending on the dataset used. In one dataset, the former method yielded superior results, while in another dataset, the latter method performed better. Although employing a fixed centroid can reduce communication cost overhead in specific scenarios, it raises the question of why the model exclusively considers the fixed centroid approach from a performance perspective.

**Questions:**

The author conducted a convergence analysis of FAFL, which seemed like an additional component of the paper. It would be valuable to directly analyze the convergence of the suggested method and provide additional supporting evidence.

---

### Official Review · Reviewer_kTy4 · 2023-11-04

**Soundness:** 3 good
**Presentation:** 3 good
**Contribution:** 2 fair
**Rating:** 5
**Confidence:** 4

**Summary:**

In this work, the authors propose a simplified version of feature alignment for federated learning
(FAFL), called SimFAFL. In particular, SimFAFL decouples the constantly updated FAFL constraints
from explicit categorical dependencies using two modular constraints, including a fixed reference
distribution and globally shared task parameters. The performance of SimFAFL is justified both
theoretically and experimentally.

**Strengths:**

1. Originality and Significance: The key idea of decoupling the constantly updated alignment
constraints into more flexible restrictions based on dual relaxations is novel. The proposed
design alleviates the problem of existing FAFL approaches, in terms of enhanced task
extensibility and reduced resource requirements for communication and computation.
2. Quality: The convergency of the proposed design is analyzed in the strongly convex case. The
evaluation incorporates the datasets of EMINST-L, FMNIST, CIFAR 10, and CIFAR 100, the
model of 6-layer CNN, and 10 existing baselines.
3. Clarity: The paper is well-organized and clearly written.

**Weaknesses:**

1. Some practical settings of federated learning are not considered in theoretical analysis. In
particular, the convergency analysis does not cover the non-convex case; and the effect of
partial client participation is not analyzed theoretically.
2. The datasets and the model for evaluation are too simple. Specifically, the chosen datasets
cannot reveal the natural non-iid data distributions of clients in federated learning, thereby
not practically demonstrating the considered problem setting.
3. From Table 2, the improvement of the proposed SimFAFL over existing baselines is marginal
or even underperforms for the four toy datasets.

**Questions:**

Why is the test accuracy of Federated Averaging (FedAvg) lower than 30% on CIFAR 100?

---

### Official Review · Reviewer_7zfg · 2023-11-07

**Soundness:** 2 fair
**Presentation:** 1 poor
**Contribution:** 2 fair
**Rating:** 3
**Confidence:** 3

**Summary:**

This paper aims to improve over existing methods for feature alignment in federated learning. The paper claims to provide a rigorous convergence analysis of FAFL. And it further proposes a framework called Sim-FAFL that mainly introduces two constraints that limit the feature distribution in local training. Comprehensive experimental assessments are conducted to validate the enhanced performance and robustness of SimFAFL.

**Strengths:**

- The problem this paper studies is very interesting and important.
- The paper's logic flow is clear. It is written in a way that first rigorously provides some theoretical guarantee, and motivated by this theory, proposes some new method.

**Weaknesses:**

Disclaimer: the reviewer is more familiar with the optimization side of federated learning, but not familiar with the state of the art in feature alignment. Thus, my view may not fully and accurately assess the actual technical contribution of this paper.

- I think the presentation of this paper requires some major revision. For example, table 1 in introduction has very limited context in introduction, and it is hard to understand the exact meaning until we get to Sec 4.

And more importantly, there is a lot of unclearness in Sec 4 when the authors present their main contribution. For example, it keeps saying "we employ FedCR" as some backbone, and never presented the detailed architecture. It poses much difficulty if not constantly referring to references. And Sec 4 mostly relies on pure words, which contain a lot of vagueness in how it is exactly done, without illustrative figure or mathematical formulation. In general, Sec 4 seems like a bit hasty, and lacking many critical details.

- the theory part, which the authors claim to be one of the main contributions, also needs some revision. As the notation is a bit sloppy, and critical inequality is given without derivation.

For example, why (3) holds is not clear to readers in current context. What is $L$ in (3) is not defined. and in the next paragraph, what is $\mu$, what is $D$, and why (4) holds is also not clear. If it is some theorem, then it needs to be presented in some formal theorem way.

And equation (5) also has similar problem, why $\eta$ suddently loses the subscript $t$? what is $E$ and $e$?

Basically, it needs to be presented more formally, clearly indicating, what is theorem, what is known in existing literature, what is derived in this paper, and clearly define key notations before using them. Otherwise, I feel very difficult to determine the correctness or usefulness of this part.

**Questions:**

Please see weaknesses.

---

### Official Review · Reviewer_NjjA · 2023-11-12

**Soundness:** 2 fair
**Presentation:** 2 fair
**Contribution:** 2 fair
**Rating:** 3
**Confidence:** 4

**Summary:**

This paper considered the problem of feature alignment for federated learning (FAFL) to address data heterogeneity in federated learning (FL). To address the limitations of existing FAFL methods in terms of i) lacking expandability and extensibility and ii) requiring additional communication and computational cost, the authors proposed a simplified version of FAFL called simFAFL. The main idea of simFAFL is to decouple the constantly updated FAFL constraints from explicit categorical dependencies. The authors conducted a theoretical analysis for their proposed algorithm's convergence rate performance. Numerical experiments are also conducted to illustrate the performance of the proposed method.

**Strengths:**

1. The idea of simplifying FAFL is interesting.

2. The authors conducted extensive experiments.

**Weaknesses:**

1. The novelty of theoretical analysis is weak.

2. The writing quality of this paper is poor and the paper is full of various types of error.

Please see the detailed comments below.

**Questions:**

1. Although the authors claimed that they are the first to conduct theoretical convergence analysis for FAFL, the novelty of the theoretical analysis is very weak. All the theoretical analysis results in Eqs. (3)-(5) and Appendix A.1 directly follow from Refs. [Li et al. 2019] and [Tan et al. 2022]. There is nothing new in the theoretical derivation. Moreover, they only considered the strongly convex setting (cf. Assumption 2) in this paper, which is rather uninteresting in practice. The bounded gradient norm assumption (Assumption 4) is also restricted and not needed in the recent literature of optimization for machine learning.

2. The writing quality of this is very poor. The paper has a lot of typos and many sentences are not proper English language. The organization of this paper is also quite hard to follow. The theoretical results presentation is quite unclear, and many mathematical notations and equations made no sense (e.g., Eq. (27). Also, the proof on Page 12 is poorly written and confusing. The authors didn't do a good job in proofreading their paper before submission and this paper is not in a ready shape.